# Ampicillin/Sulbactam Treatment Modulates NMDA Receptor NR2B Subunit and Attenuates Neuroinflammation and Alcohol Intake in Male High Alcohol Drinking Rats

**DOI:** 10.3390/biom10071030

**Published:** 2020-07-10

**Authors:** Fawaz Alasmari, Hasan Alhaddad, Woonyen Wong, Richard L. Bell, Youssef Sari

**Affiliations:** 1Department of Pharmacology and Toxicology, College of Pharmacy, King Saud University, Riyadh 11451, Saudi Arabia; ffalasmari@ksu.edu.sa; 2Department of Pharmacology and Experimental Therapeutics, University of Toledo, College of Pharmacy and Pharmaceutical Sciences, Toledo, OH 43614, USA; hasan.alhaddad@rockets.utoledo.edu (H.A.); woonyen.wong@rockets.utoledo.edu (W.W.); 3Department of Psychiatry and Institute of Psychiatric Research, Indiana University School of Medicine, Indianapolis, IN 46202, USA

**Keywords:** ethanol, AMP/SUL, GLT-1, NMDA, neuroinflammation

## Abstract

Exposure to ethanol commonly manifests neuroinflammation. Beta (β)-lactam antibiotics attenuate ethanol drinking through upregulation of astroglial glutamate transporters, especially glutamate transporter-1 (GLT-1), in the mesocorticolimbic brain regions, including the nucleus accumbens (Acb). However, the effect of β-lactam antibiotics on neuroinflammation in animals chronically exposed to ethanol has not been fully investigated. In this study, we evaluated the effects of ampicillin/sulbactam (AMP/SUL, 100 and 200 mg/kg, i.p.) on ethanol consumption in high alcohol drinking (HAD1) rats. Additionally, we investigated the effects of AMP/SUL on GLT-1 and *N*-methyl-d-aspartate (NMDA) receptor subtypes (NR2A and NR2B) in the Acb core (AcbCo) and Acb shell (AcbSh). We found that AMP/SUL at both doses attenuated ethanol consumption and restored ethanol-decreased GLT-1 and NR2B expression in the AcbSh and AcbCo, respectively. Moreover, AMP/SUL (200 mg/kg, i.p.) reduced ethanol-increased high mobility group box 1 (HMGB1) and receptor for advanced glycation end-products (RAGE) expression in the AcbSh. Moreover, both doses of AMP/SUL attenuated ethanol-elevated tumor necrosis factor-alpha (TNF-α) in the AcbSh. Our results suggest that AMP/SUL attenuates ethanol drinking and modulates NMDA receptor NR2B subunits and HMGB1-associated pathways.

## 1. Introduction

The World Health Organization reported that consumption of alcoholic beverages has increased significantly in recent years and often leads to negative health consequences [1]. Our previous studies revealed that chronic ethanol consumption decreased the expression of astro glial glutamate transporters and increased extracellular glutamate concentrations in the nucleus accumbens (Acb) [2,3,4]. The elevation of extracellular glutamate concentrations has been found to enhance ethanol consumption in mice [5]. It is noteworthy that Acb is a central brain region that regulates ethanol consumption and relapse-like behaviors [6,7]. Acb is composed of two subregions; Acb core (AcbCo) and Acb shell (AcbSh) and that chronic ethanol intake and withdrawal may cause differential effects on the expression of glutamatergic receptors in these subregions [8]. It has been suggested that cue and context-induced relapse to ethanol seeking behaviors are critically regulated by AcbCo and AcbSh, respectively [9].

Studies from ours and others have found that the increases in extracellular glutamate concentrations in the Acb is mediated mainly by reduced glutamate transporter-1 (GLT-1) expression [3,4,10]. It is important to note that β-lactam antibiotics (e.g., ceftriaxone and ampicillin, AMP) increased GLT-1 expression in the Acb and attenuated ethanol-drinking behaviors [11,12,13,14]. Additionally, a β-lactamase inhibitor, clavulanic acid, decreased ethanol and cocaine-seeking behaviors, and increased the expression of GLT-1 in the Acb [15,16]. The increase in the synaptic glutamate concentrations is associated with alteration in certain glutamate receptors expression [17] as well as neuroinflammation [18].

The role of ionotropic glutamate receptors, including *N*-methyl-d-aspartate receptors (NMDAR), in ethanol seeking and drinking behaviors, has been investigated extensively [19,20,21]. Treatment with memantine, an NMDAR antagonist, decreased ethanol self-administration in animals [22]. Additionally, treatment with MRZ 2/579, an NMDAR antagonist, reduced ethanol withdrawal symptoms in rats [21]. Studies have shown that acute ethanol consumption inhibits the function of NMDARs [23], while chronic ethanol consumption associated with increased NMDARs activity [24,25,26]. Moreover, exposure to ethanol induced changes in the expression of NMDAR subunits, which depend on the brain region, age of animal, and ethanol exposure paradigm [27,28]. There are several isoforms of NMDARs, all of which are highly expressed in the brain [29]. Among these subunits, NR2A and NR2B exhibited the higher sensitivity toward the inhibitory effects of ethanol compared to other NMDA receptor subunits [30]. Additionally, studies showed that stimulation of NMDA receptors was associated with neuroinflammation [31]. Moreover, NMDA exposure increased the release and expression of high mobility group box 1 (HMGB1) and that a NR2B inhibitor attenuates these effects [32]. Furthermore, treatment with memantine, NMDAR antagonist, attenuated colchicine-induced neuroinflammation [33].

The role of metabotropic glutamate receptors (mGluRs), particularly mGluR5, in the attenuation of ethanol seeking behaviors is well documented, and these receptors are therapeutically targeted for the treatment of alcohol use disorder (reviewed in [34,35]). The blockade of mGluR5 reduced ethanol self-administration and relapse-like ethanol seeking behaviors [36]. The long-term depression in corticostriatal synapses was attenuated with treatments of mGluR5 antagonist indicating the involvement in the striatal synaptic plasticity [37]. The availability of mGluR5 in the mesocorticoimbic system was altered with ethanol exposure [38,39]. Stimulation of mGluR5 was associated with anti-inflammatory effects, in part, through reduction of microglial activation [40]. 

HMGB1 can act as an inflammatory mediator and is known to stimulate certain receptors, including receptor for advanced glycation end-products (RAGE) receptor [41]. This receptor, in turn, activates nuclear factor kappa-B (NF-κB) and a host of other intracellular cascades [41], leading to an increase in the expression of neuroinflammatory markers/mediators, such as tumor necrosis factor-alpha (TNF-α) and interleukin 1-beta (IL-1β) [42]. Exposure to ethanol induced neuroinflammation by modulating signaling pathways [43,44,45], for example by elevating the expression of neuroinflammatory mediators (e.g., TNF-α) in the brain [46]. The gene and protein expression of HMGB1 were increased in hippocampal-entorhinal cortical cell treated with ethanol, an effect associated with increased gene expression of TNF-α and IL-1β [47]. In clinical studies, elevation of HMGB1 levels has been observed in alcohol binge female drinkers [48]. Additionally, postmortem human alcoholic hippocampus showed an increase in the HMGB1expression [49]. HMGB1 is implicated in the development of ethanol withdrawal-induced neurotoxicity, a process known to involve glutamate overstimulation [50]. It is important to note that HMGB1 release is also required for glutamate toxicity in the absence of ethanol [32].

Collectively, these studies suggest that NMDAR and HMGB1 pathways are crucial for the development of ethanol drinking and withdrawal behaviors, and associated neuroinflammation. Therefore, we hypothesized, in this study, that chronic ethanol consumption may induce alterations on NMDAR subunits, mGluR5 and inflammatory mediators in the AcbCo and AcbSh, and that ampicillin/sulbactam (AMP/SUL) treatment, a compound that has both β-lactam antibiotic and β–lactamase inhibitor pharmacological effects, would modulate these changes. It is important to note that both AMP and SUL can cross the blood brain barrier and can have a therapeutic effect against bacterial infection, which is the cause of meningitis [51]. It is noteworthy that AMP/SUL treatment has been shown to attenuate reinstatement to cocaine seeking behaviors, in part by upregulating astroglial glutamate transporters in the AcbCo and AcbSh [52]. To evaluate this hypothesis, we first investigated the effect of AMP/SUL treatment on chronic ethanol consumption in high alcohol drinking (HAD1) rats. Then, we measured the protein expression of GLT-1, NR2A and NR2B subunits, mGluR5, HMGB1, RAGE, and TNF-α in the AcbCo and AcbSh. Our results showed that chronic ethanol consumption induces inflammatory response in the AcbSh, and that AMP/SUL treatment attenuates these effects. We have previously found that exposure to AMP/SUL (200 mg/kg, i.p.) alone did not induce or reduce conditional place preference using alcohol preferring (P) rats in pre- and post-conditioning tests [52,53]. However, AMP did upregulate GLT-1 in cultured cortical neurons [54]. Therefore, in our study, we hypothesized that AMP/SUL reversed the downregulatory effects of ethanol on GLT-1 expression. Hence, AMP/SUL might attenuate the resulting deleterious effects, including neuroinflammation.

## 2. Materials and Methods

### 2.1. Animals and Drinking Protocol

Male HAD1 rats, which serve as an animal model of alcohol use disorders [55,56,57,58], were used in the present study. Rats were housed in a vivarium that is fully certified by the Association for Assessment and Accreditation of Laboratory Animal Care International and is maintained at ~50% humidity and ~23 °C on a 12 h/12 h reverse dark/light cycle (dark onset at 10:00 h). All procedures were approved by the Indiana University School of Medicine Institutional Animal Care and Use Committee and are in accordance with the National Institutes of Health animal care directives (e.g., The Guide for the Care and Use of Laboratory Animals). Animals were housed in hanging stainless steel wire mesh cages, with a Plexiglas^®^ shelf located in the bottom of the cage so that they were not continuously in contact with the wire mesh floor. Rats had ad libitum access to food throughout the procedures. Rats started the experimental drinking procedure at age of 55 days. Rats had nine weeks of exposure to continuous three-bottle choice drinking, with two experimental procedures. We measured the average ethanol intake during the last three weeks, which was considered as the baseline. We used our established criteria for selection of rats that developed dependence to ethanol. Rats developed dependence to ethanol by five weeks under the 15% and 30% *v*/*v* ethanol free choice drinking schedule. Rats had to drink at least 4 g/kg/day or higher in order to be kept in the study. The first procedure was exposing groups of rats to a water bottle and two bottles of ethanol (15% and 30%, *v*/*v*, available concurrently) for nine weeks which represented ethanol-control, AMP/SUL (100 mg/kg, i.p.) and AMP/SUL (200 mg/kg, i.p.) groups, whereas the second experimental procedure was exposing group of rats to three bottles of water for nine weeks which represented the water-control (ethanol-naïve) group. The positions of each bottle were semi-randomized on an every-other-day or every third day basis. Immediately after the rats drank for eight weeks, two doses (100 or 200 mg/kg, i.p.) of AMP/SUL were tested during the 9th week of drinking. There were two control groups, which received equivolume doses of saline/vehicle. A new solution of AMP/SUL was mixed each day. AMP/SUL was injected intraperitoneally 30-min before the dark cycle. These injections took place across five consecutive days, during the 9th week of ethanol or water drinking. One hour before dark onset, animal body weights, as well as food hopper and bottle weights, were taken on Monday through to Friday. Food hoppers and fluid bottles were topped off and placed on their respective cages at dark onset (10:00 a.m.). Twenty-four hours after the last AMP/SUL i.p. injection, rats were quickly euthanized with carbon dioxide followed by decapitation, with the brains harvested and placed in isopentane on dry ice and stored at −80 °C for subsequent protein level analyses. The AMP/SUL doses were selected based on previous studies showing that ceftriaxone, a β-lactam antibiotic, could reduce ethanol intake and consistently affect protein levels in the mesocorticolimbic reward circuit at 100 mg/kg or 200 mg/kg, i.p. [13,14]. Similar findings have been observed with AMP (100 mg/kg, i.p.) and AMP/SUL (100 mg/kg or 200 mg/kg, i.p.) in previous studies [11,59].

### 2.2. Brain Dissection

The micro-punch procedure was used to isolate Acb subregions (AcbCo and AcbSh) using a cryostat machine maintained at −20 °C in order to keep the samples frozen. The Paxinos and Watson Atlas [60] was used to evaluate the location and thickness to isolate these brain regions.

### 2.3. Western Blot Analyses

Immunoblot assays were performed to investigate changes in the expression of GLT-1, NR2A, NR2B, HMGB1, RAGE, TNF-α, and β-tubulin in the AcbCo and AcbSh as described in previous studies [52,61]. A lysis buffer containing protease inhibitor was used during the homogenization of AcbCo and AcbSh samples. Detergent compatible protein assay (Bio-Rad, Hercules, CA, USA) was then used to quantify the amount of protein in each tissue sample. An equal amount of protein from each sample was mixed with laemmli dye, and the mixtures were loaded onto polyacrylamide gels (8–10%). Proteins were then transferred from the gels to a polyvinylidene difluoride (PVDF) membrane. Subsequently, the PVDF membranes were blocked with 3–5% fat free milk in Tris-buffered saline with Tween-20 (TBST) at room temperature for one hour. The appropriate primary antibodies were incubated with the membranes at 4 °C (overnight): guinea pig anti-GLT-1 (1:5000, Abcam, Cambridge, UK), rabbit anti-NR2A antibody (1:500; EMD Millipore, Burlington, MA, USA), rabbit anti-NR2B antibody (1:500; EMD Millipore), rabbit anti-HMGB1 (1:1000; Abcam), rabbit anti-RAGE (1:1000; Abcam), and rabbit anti-TNF-α (1:500; Abcam). Mouse anti-β-tubulin (1:1000; BioLegend, San Diego, CA, USA) was used as a control loading protein. On the following day, membranes were washed five times with TBST followed by blocking with 3% fat free milk for 30 min at room temperature. Membranes were then incubated with secondary antibody (1:3000) for 90 min. Subsequently, membranes were washed five times with TBST. Chemiluminescent reagents (Super Signal West Pico, Pierce Inc., Appleton, WI, USA) were incubated with the dried membranes for protein detection. Membranes were developed on a GeneSys imaging system and digitized blot images were developed. Quantification and analysis of the expression of GLT-1, NR2A, NR2B, HMGB1, RAGE, TNF-α, and β-tubulin blots were performed using ImageJ software (v1.53a). Water-control group data were represented as 100% (relative to water-control) to assess any changes in the expression of protein of interest in the AcbCo and AcbSh as described in our previous studies [62,63,64,65,66,67]. Therefore, in the same gel run, the expression of the protein in each group was normalized to the average values of the water-control group.

### 2.4. Statistical Analyses

#### 2.4.1. Drinking-Solution Data

All statistical analyses were conducted with Graphpad Prism software (v5.04), with alpha set at *p* < 0.05. A two-way (or mixed model) ANOVA was conducted on 24 h ethanol intake, 24 h water intake, 24 h food intake, and daily body weight during the five days of AMP/SUL treatment. Significant interactions and/or main effects were decomposed into simple main effects and, where the latter were significant, a priori Dunnett’s *t*-test multiple comparisons were conducted for each dose vs. the ethanol-saline control group.

#### 2.4.2. Western Blot Data

All statistical analyses were performed using Graphpad Prism software. A one-way ANOVA with Newman-Keuls as a post-hoc multiple comparison test was used to analyze the Western blot data as percentage (relative to water-control values) ratio to the loading protein, β-tubulin. The data are reported for a *p* < 0.05 level of significance.

## 3. Results

### 3.1. Effects of AMP/SUL (0, 100 or 200 mg/kg) on Fluid and Food Intake as Well as Body Weight

#### 3.1.1. Effects of AMP/SUL (0 or 100 or 200 mg/kg) on 24 h Ethanol Intake (g/kg/day)

Statistical analysis of the ethanol drinking data revealed a significant Dose by Day interaction (*p* < 0.001). Each Day simple main effect was significant from Days 1–5, with Dunnett’s t-tests showing that significant decrease in ethanol intake on Day 1 (*p* < 0.001) and Days 2–5 (*p* < 0.0001), as shown in Figure 1A (*n* = 9/group).

#### 3.1.2. Effects of AMP/SUL (0 or 100 or 200 mg/kg) on 24 h Water Intake (ml/day)

Statistical analysis of water intake revealed significant Dose by Day interaction (*p* < 0.01). Dunnett’s t-test revealed significant increase in water intake with both doses (100 mg/kg and 200 mg/kg) relative to ethanol-vehicle group (Day 1, *p* < 0.01; Days 2 through 4, *p* < 0.001 with 100 mg/kg AMP/SUL and *p* < 0.0001 with 200 mg/kg AMP/SUL; Days 5, *p* < 0.001), as shown in Figure 1B (*n* = 9/group).

#### 3.1.3. Effects of AMP/SUL (0 or 100 or 200 mg/kg) on Average Body Weight (mL/day)

Using two-way (mixed) ANOVA, average body weight was analyzed as well. The Dose by Day interaction and the main effect of Dose or Days were not significant (*p* > 0.05), as shown in Figure 1C (*n* = 9/group).

#### 3.1.4. Effects of AMP/SUL (0 or 100 or 200 mg/kg) on Food Intake (g/day)

Food intake data across the first 4 days of testing was analyzed (data from the 5th test day was corrupted). Statistical analysis of food intake revealed not significant Dose by Day interaction (*p* > 0.05). The main effect of Dose (*p* < 0.001) and Day (*p* < 0.0001) were significant. The Dunnett’s *t*-test revealed that both doses significantly reduced food intake only on Day 1 with 100 mg/kg AMP/SUL (*p* < 0.0001) and 200 mg/kg AMP/SUL (*p* < 0.001), as shown in Figure 1D (*n* = 9/group).

### 3.2. Effects of AMP/SUL (0 or 100 or 200 mg/kg) on the Expression of GLT-1 in the AcbCo and AcbSh of Chronically Ethanol Drinking HAD1 Rats

One-way ANOVA revealed no significant difference in GLT-1 expression levels in the AcbCo, among the water-control, ethanol-control, and ethanol treated with 100 or 200 mg/kg (i.p.) AMP-SUL groups AcbCo (*p* > 0.05, (Figure 2B), *n* = 5/group). However, there was a significant difference in GLT-1 expression among the four groups in the AcbSh (*F* (3, 16) = 22.53, *p* < 0.0001, (Figure 2D), *n* = 5/group). Newman-Keuls multiple comparisons post-hoc analysis showed a decrease in GLT-1 expression in the AcbSh of the ethanol-treated vs water control groups. Statistical analysis also showed that AMP-SUL (100 or 200 mg/kg, i.p.) increased GLT-1 expression in the AcbSh as compared to the ethanol-control group.

### 3.3. Effects of AMP/SUL (0 or 100 or 200 mg/kg) on the Expression of NR2B and NR2A in the AcbCo and AcbSh of Chronically Ethanol Drinking HAD1 Rats

We investigated the effects of chronic ethanol consumption and AMP-SUL treatments on the expression of NR2B in the AcbCo and AcbSh. There was a significant change among all four groups in the AcbCo (*F* (3, 16) = 6.781, *p* = 0.0037 (Figure 3B), *n* = 5/group). Newman-Keuls multiple comparisons post-hoc analysis revealed a reduction in the expression of NR2B in the ethanol-control group compared to water-control group, and that the ethanol-AMP-SUL groups (100 and 200 mg/kg, i.p.) showed higher NR2B expression compared to the ethanol control group. However, there was no significant difference in the expression of NR2B among all groups in the AcbSh (*p* > 0.05 (Figure 3B), *n* = 5/group). We further determined the effects of AMP/SUL treatments on the expression of NR2A and NR2B subunits in the AcbCo and AcbSh. There was no significant change in NR2A expression in the AcbCo among the water-control, ethanol-control, and ethanol-AMP-SUL (100 and 200 mg/kg, i.p.) groups (*p* > 0.05 (Figure 3D), *n* = 5/group). Similarly, there was no significant change in NR2A expression in the AcbSh among all groups (*p* > 0.05 (Figure 3D), *n* = 5/group).

### 3.4. Effects of AMP/SUL (0 or 100 or 200 mg/kg) on the Expression of HMGB1 and RAGE in the AcbCo and AcbSh of Chronically Ethanol Drinking HAD1 Rats

There was no significant change in the expression of HMGB1 in the AcbCo among all groups (*p* > 0.05 (Figure 4B), *n* = 5/group). One-way ANOVA followed by Newman-Keuls multiple comparison post-hoc analyses revealed that there was an increase in HMGB1 expression in the AcbSh of the ethanol-control group as compared to the water-control group, and this effect was attenuated by 200 mg/kg, but not 100 mg/kg, dose of AMP/SUL (*F* (3, 16) = 19.73, *p* < 0.0001 (Figure 4B), *n* = 5/group). We determined the expression of RAGE in the AcbCo and AcbSh of water-control, ethanol-control, and ethanol-AMP-SUL groups. One-way ANOVA did not reveal any difference in RAGE expression in the AcbCo among these groups (*F* (3, 16) = 0.3135, *p* = 0.8154 (Figure 4D), *n* = 5/group). However, there was a significant change in RAGE expression in the AcbSh among these groups (*F* (3, 16) = 23.92, *p* < 0.0001 (Figure 4D), *n* = 5/group). There was an increase in RAGE expression in the AcbSh after chronic ethanol consumption compared to the water control group, and this increase was attenuated by 200 mg/kg, but not 100 mg/kg, dose of AMP/SUL.

### 3.5. Effects of AMP/SUL (0 or 100 or 200 mg/kg) on the Expression of TNF-α in the AcbCo and AcbSh of Chronically Ethanol Drinking HAD1 Rats

We investigated the effects of chronic ethanol consumption and AMP-SUL treatments on the expression of TNF-α in the AcbCo and AcbSh. There was no change in the expression of TNF-α in the AcbCo among all groups (*F* (3, 16) = 2.9625, *p* = 0.0636 (Figure 5B), *n* = 5/group). However, there was significant TNF-α expression elevation in the AcbSh of the ethanol-control as compared to water-control groups. Importantly, post-hoc analyses revealed that AMP/SUL (100 or 200 mg/kg, i.p.) attenuated the effect of chronic ethanol consumption on TNF-α expression in the AcbSh (*F* (3, 16) = 23.70, *p* < 0.0001 (Figure 5D), *n* = 5/group).

### 3.6. Effects of AMP/SUL (0 or 200 mg/kg) on the Expression of mGluR5 in the AcbCo and AcbSh of Chronically Ethanol Drinking HAD1 Rats

We further determined the expression of mGluR5 in the AcbCo and AcbSh. Statistical analysis did not reveal any significant changes between water-control, ethanol-control and ethanol-treated (200 mg/kg AMP/SUL, i.p.) groups in the AcbCo (*F* (2, 12) = 3.220, *p* = 0.6863 (Figure 6B), *n* = 5/group). However, one-way ANOVA followed by Newman-Keuls post-hoc analyses revealed that mGluR5 expression was reduced in ethanol-control group compared with water-control group, and that AMP/SUL (200 mg/kg, i.p.) attenuated this effect (*F* (2, 12) = 8.625, *p* = 0.0048 (Figure 6D), *n* = 5/group) in the AcbSh.

## 4. Discussion

This study confirmed that AMP/SUL treatment attenuates ethanol drinking behavior, by extending previous findings with AMP in P rats to include HAD1 rats as another animal model of alcoholism [11]. In addition, AMP/SUL treatment attenuated the effects of chronic ethanol drinking on GLT-1, HMGB1, RAGE, and TNF-α expression in the AcbSh. AMP/SUL also attenuated ethanol-induced decreases in NR2B expression in the AcbCo compared to the ethanol-control group. The present data suggest that AMP/SUL attenuates both ethanol drinking behaviors and neuroinflammation, at least in part, through modulating glutamatergic activity.

The present findings revealed that AMP/SUL (100 mg/kg or 200 mg/kg, i.p.) attenuated ethanol drinking and normalized the expression of GLT-1 in the AcbSh. Other studies from our laboratory have found that ceftriaxone reduced ethanol drinking concurrently with an upregulation of GLT-1 expression in the Acb of P rats [12,13]. Furthermore, our laboratory reported that AMP treatment for five days attenuated ethanol drinking behavior in P rats, an effect that was associated with upregulation of GLT-1 expression in the Acb [11]. In addition, β-lactamase inhibitors, clavulanic acid and SUL, showed ability to upregulate GLT-1 expression in rats [15,16,68]. In the present study, we found that nine weeks of ethanol consumption reduced the expression of GLT-1 in the AcbSh, but not in the AcbCo, and that combination of AMP and SUL attenuated this effect. This is in agreement with previous studies from our laboratory demonstrating that ethanol drinking for five weeks resulted in downregulation of GLT-1 expression in the Acb [3,69]. We have previously reported that chronic ethanol drinking for approximately 7–8 weeks decreased GLT-1 expression in the AcbSh, but not in the AcbCo of P rats [70]. The present study and previous studies from our laboratory revealed that the increase in water intake might be a compensatory response for the attenuation of ethanol intake during treatments with β-lactam antibiotics [13,14]. At Day 1 of treatment, there was a significant reduction in food intake in AMP/SUL-treated groups compared with ethanol-control group. We believe that this reduction of food intake due to common side effects of AMP/SUL i.p. injection. Taken together, these studies suggest that chronic ethanol consumption downregulates GLT-1 expression in the AcbSh, which may lead to an increase in extracellular concentrations of glutamate in this region of the mesocorticolimbic reward circuit [3].

In the present study, nine weeks of ethanol drinking decreased NR2B expression in the AcbCo, but not in the AcbSh. Importantly, previous reports revealed that chronic ethanol exposure modulated the expression of NR2B in the frontal cortex and hippocampus [28]. For example, chronic ethanol consumption increased NR2B expression in the hippocampus in adult Wistar rats, whereas the expression of NR2B was downregulated in the frontal cortex [28]. In addition, adult and adolescent rats exhibited different patterns of NR2B expression after chronic ethanol exposure, whether they examined 24 h or 2 weeks of ethanol withdrawal [27,28]. Moreover, chronic stress also reduced cell membrane expression of NR2B in the Acb [71]. Furthermore, chronic stress decreased NR2B expression in the prefrontal cortex of male Sprague–Dawley rats [72]. It appears that multiple factors such as brain region, ethanol and/or stress exposure paradigm, as well as rat line used can affect alterations of NR2B expression in the mesocorticolimbic system. Our results in this study showed that ethanol-induced reduction in NR2B was observed in the AcbCo, and AMP/SUL treatment attenuated this effect, although another study revealed that 6 month exposure to free choice 15% and 30% of ethanol induced an increase in NR2B expression in the AcbCo and central nucleus of amygdala (CeA) of P rats [8]. The alteration of NR2B expression might be a neuroadapative mechanism for the effects of chronic ethanol exposure on glutamate homeostasis. Since NR2B expression was not affected in the AcbSh but decreased in the AcbCo, we hypothesized in this study that high extracellular glutamate concentrations in the AcbSh may activate NR2B subunits of NMDA receptor. This effect may lead to upregulatory effects on neuroinflammatory factors.

Differential effects of ethanol drinking on NR2A expression in the AcbCo, AcbSh and CeA of P rats have been reported, which were dependent on types of ethanol access (intermittent vs continuous) and lengths of ethanol withdrawal (24 h vs 4 weeks) [8]. Other studies also reported that ethanol exposure affected NR2A expression in the hippocampus and frontal cortex of Wistar rats [27,28]. However, the present study did not demonstrate any marked changes in the expression of NR2A in either the AcbCo or AcbSh of HAD1 rats. It is noteworthy that the P rat was derived from a closed colony of Wistar rats [55,56,57]. This is important since HAD1 rats were derived from an eight inbred rat strain intercross, which included non Wistar inbred rat strains [55,56,57]. Thus, genetics may play a part in the diversity of findings. Nevertheless, for HAD1 rats, it appears that the NR2A subunit is not affected by ethanol drinking, at least under the present study’s conditions.

Along with the NMDAR and GLT-1 alterations, we found that the neuroinflammatory signaling pathway, which include HMGB1, RAGE and TNF-α, is also altered in the AcbSh of HAD1 rats. Our results showed that chronic ethanol drinking increased the expression of HMGB1, RAGE and TNF-α, while concurrently reducing GLT-1 expression in the AcbSh. Studies demonstrated that binge drinking of ethanol for 10 days followed by abstinence period of 27 h increased mRNA and protein expression of HMGB1 in the brain of C57BL/6 male mice, an effect associated with upregulated TNF-α in the brain, but not in the serum [73]. We showed here that chronic ethanol consumption increased the expression of TNF-α and HMGB1 in the AcbSh, but not in the AcbCo. The upregulatory effect of ethanol on TNF-α level was also found in the prefrontal cortex in adolescent female mice following intermittent treatments of ethanol for two weeks [74]. Studies suggested that the blood level of TNF-α is elevated for 3 h after the last treatment of ethanol, while the levels of TNF-α in the brain remain upregulated for longer time [73]. Additionally, the present study showed that RAGE, a HMGB1 receptor, expression was increased in the ACbSh in ethanol control group. It is important to note that mRNA expression of RAGE and HMGB1 as well as acetylated and phosphorylated HMGB1 (but not total HMGB1) protein levels were elevated in the cerebellum of C57BL/6J female mice after 5-week ethanol exposure [75]. Nevertheless, AMP/SUL treatments (200 mg/kg) normalized the levels of TNF-α, RAGE and HMGB1, suggesting an interaction between the activation of neuroinflammation and the dysregulation of glutamatergic system in the Acb in relation to chronic ethanol drinking. Interestingly, a previous study found excitotoxicity induced by glutamate/NMDA injections, which was able to increase media acetyl-HMGB1 in a bath of hippocampal-entorhinal cortex cells, and this is consistent with neuronal secreted HMGB1 [76]. This latter study also reported that this effect was abolished after treatment with ifenprodil, an NR2B inhibitor. Another study reported that an NR2B antagonist (Ro25-6981), but not an NR2A antagonist (NVP-AAM077), was able to attenuate HMGB1 expression in cortical neurons treated with NMDA [77]. Thus, glutamate might induce neuroinflammation through stimulation of post-synaptic glutamate receptors, in particular those including NR2B subunits. In addition, GLT-1 upregulation by AMP/SUL treatments (200 mg/kg) was associated with reduced HMGB1, RAGE and TNF-α in the AcbSh, which suggests that restoring extracellular glutamate concentration can attenuate both ethanol drinking and neuroinflammation. However, AMP/SUL (100 mg/kg, i.p.) attenuated only TNF-α expression but there were non-significant reductions on both HMGB1 and RAGE. It is important to consider that AMP/SUL (200 mg/kg, i.p.) group had higher GLT-1 expression in the AcbSh compared to AMP/SUL (100 mg/kg, i.p.) group. These findings are in agreement with a previous study showing that ceftriaxone treatment attenuated neuroinflammation through modulating glutamatergic pathways in an animal model of Parkinson disease [18]. Alternatively, studies suggest that ethanol has the ability to allow the transfer gut microbial components into the circulatory system, which may lead to neuroinflammation through gut-brain interplay [76,78]. This pathway was targeted with oral treatments of cocktail antibiotic containing 100 mg/kg AMP, which could attenuate the load of bacteria in the intestine and associated neuroinflammation in C57BL/6J female mice [76]. This raises the possibility that AMP/SUL treatments might attenuate neuroinflammation via gut-brain pathway. Taken together, β-lactam antibiotics might attenuate RAGE and TNF-α expression through normalization of HMGB1 inflammatory pathway, protecting gut microbiome, and modulating glutamatergic pathways.

We further determined the effects of nine-week ethanol drinking and treatments with AMP/SUL (200 mg/kg, i.p.) on the protein expression of mGluR5 in the AcbCo and AcbSh. We found that mGluR5 expression was decreased in the AcbSh in the ethanol-control group compared with ethanol naïve group. AMP/SUL treatments attenuated this effect. Interestingly, activation of mGluR5 exerts anti-inflammatory properties through reduction in nitric oxide, TNF-α and reactive oxygen species production in vitro [40]. Moreover, reduction of the protein and mRNA expression of inflammatory cytokines, IL-1b, IL6 and TNF-α were observed after treatments with mGluR5 agonists (VU0360172 or CHBG) [79]. Our data suggest that there is interaction between mGluR5 and neuroinflammatory proteins, and that AMP/SUL modulates these pathways in rats chronically exposed to ethanol.

## 5. Conclusions

Overall, the present study reports that chronic ethanol consumption downregulated GLT-1 expression in the AcbSh, but not AcbCo, of HAD1 rats. This would, in turn, increase extracellular glutamate concentrations in the AcbSh. As NR2B expression was reduced in the AcbCo but unchanged in the AcbSh, high glutamate concentrations in the AcbSh may further stimulate NR2B subunits-containing NMDA receptors. This overstimulation of NR2B in the AcbSh might also be associated with activated neuroinflammatory parameters (HMGB1, RAGE and TNF-α) in the AcbSh. Therefore, AMP/SUL treatments countered chronic ethanol-induced downregulation of GLT-1, as well as NR2B expression in the AcbCo. At the same time, AMP/SUL treatments reversed ethanol-induced upregulation of neuroinflammatory markers in the AcbSh of HAD1 rats. In addition, our data revealed that the downregulatory effects of ethanol on the mGluR5 were associated with upregulation of HMGB1, RAGE and TNF-α levels in the AcbSh, and AMP/SUL modulated these pathways, which further suggests the anti-inflammatory properties of mGluR5. However, the causal link and crosstalk between excitotoxicity and neuroinflammation or between ethanol consumption and neuroinflammation need further validation. Moreover, future studies are warranted to investigate the neuronal or cell type involving these excitatory and neuroinflammatory target proteins using immunocytochemistry and stereological system. Since our study was conducted using male rat model, future studies are warranted to verify these findings in female animal models. The present findings indicate that chronic ethanol exposure modulates glutamatergic and neuroinflammatory activity, and that β-lactam antibiotics (i.e., AMP) and β-lactamase inhibitors (i.e., SUL) have the potential to restore homeostasis in these systems.

## Figures and Tables

**Figure 1 biomolecules-10-01030-f001:**
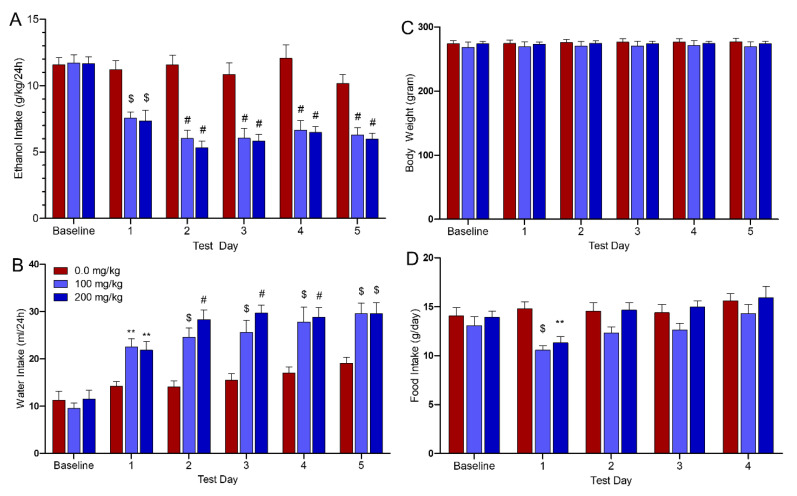
Effects of ampicillin/sulbactam (AMP/SUL) treatments (100 mg/kg and 200 mg/kg, i.p.) for five consecutive days on (**A**) Ethanol consumption (g/kg of average body weight/day), (**B**) Water intake (mL/day), (**C**) Body weight, and (**D**) Food intake. Statistical analyses revealed that AMP/SUL consistently reduced ethanol intake with a concomitant increase in water intake. However, there was an increase in water intake on Day 5 as compared to baseline (*p* < 0.01) in ethanol-control group. While food intake was transiently reduced on the 1st day of treatment, there were no other significant effects of AMP/SUL. In addition, AMP/SUL did not affect body weight. The values are expressed as mean ± SEM (*n* = 9/group), (* *p* < 0.05 and ** *p* < 0.01, $ *p* < 0.001 and # *p* < 0.0001).

**Figure 2 biomolecules-10-01030-f002:**
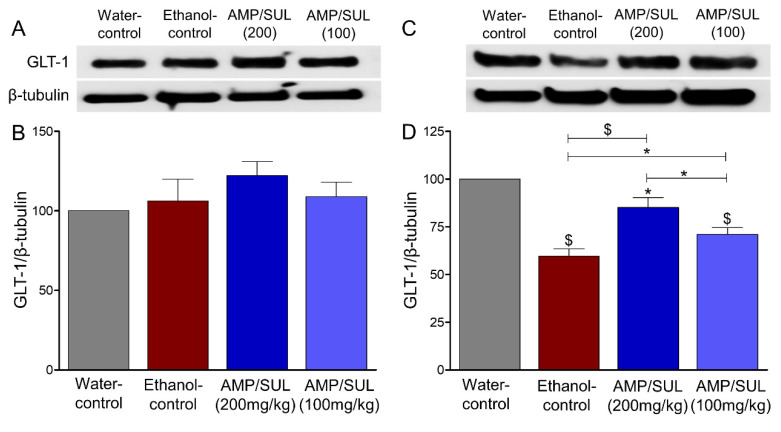
Effects of AMP/SUL treatments (100 mg/kg and 200 mg/kg, i.p.) for five days on the expression of GLT-1 in the AcbCo and AcbSh. (**A**) Immunoblot of GLT-1 and β-tubulin in the AcbCo. (**B**) Quantitative analysis using one-way ANOVA followed by Newman-Keuls test indicated that there were no significant differences in the expression of GLT-1 among water-control, ethanol-control, ethanol-AMP/SUL (100 mg/kg), and ethanol-AMP/SUL (200 mg/kg) in the AcbCo. (**C**) Immunoblot of GLT-1 and β-tubulin in the AcbSh. (**D**) Quantitative analysis using one-way ANOVA followed by Newman-Keuls test indicated that there was a significant decrease in GLT-1 expression in the ethanol control group compared to water-control group, while post-treatments with AMP/SUL (100 and 200 mg/kg) upregulated GLT-1 expression compared to ethanol-control group in the AcbSh. Ethanol-AMP/SUL (100 and 200 mg/kg) groups had lower GLT-1 expression compared to water-control group. The symbol of statistical significance between any group and water-control group is shown on the bar of the group. Water-control group data were represented as 100% (relative to water-control). The values are expressed as mean ± SEM (*n* = 5/group), (* *p* < 0.05, ** *p* < 0.01 and $ *p* < 0.001).

**Figure 3 biomolecules-10-01030-f003:**
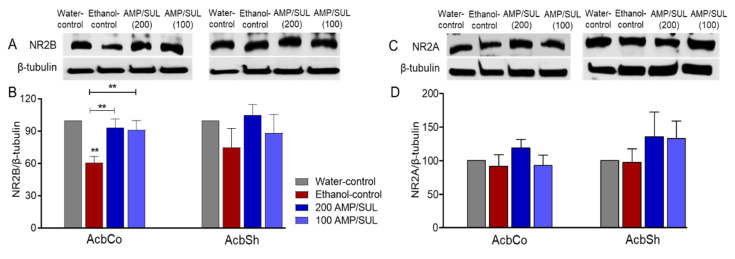
Effects of AMP/SUL treatments (100 mg/kg and 200 mg/kg, i.p.) for five days on the expression of NR2B and NR2A in the AcbCo and AcbSh. (**A**) Immunoblot of NR2B and β-tubulin in the AcbCo and AcbSh. (**B**) Quantitative analysis using one-way ANOVA followed by Newman-Keuls test indicated that there was a significant decrease in the NR2B expression in the ethanol-control group compared to the water-control group, while post-treatments with AMP/SUL (100 and 200 mg/kg) for five days normalized NR2B expression in the AcbCo. Quantitative analysis using one-way ANOVA followed by Newman-Keuls test indicated that there were no significant differences in the expression of NR2B among the water-control, ethanol-control, ethanol-AMP/SUL (100 mg/kg), and ethanol-AMP/SUL (200 mg/kg) in the AcbSh. (**C**) Immunoblot of NR2A and β-tubulin in the AcbCo and AcbSh. (**D**) Quantitative analysis using one-way ANOVA followed by Newman-Keuls test indicated that there were no significant differences in the expression of NR2A between water-control, ethanol-control, ethanol-AMP/SUL (100 mg/kg), and ethanol-AMP/SUL (200 mg/kg) in the AcbCo. Quantitative analysis using one-way ANOVA followed by Newman-Keuls test indicated that there were also no significant differences in the expression of NR2A among water-control, ethanol-control, ethanol-AMP/SUL (100 mg/kg), and ethanol-AMP/SUL (200 mg/kg) rats in the AcbSh. The symbol of statistical significance between any group and water-control group is shown on the bar of the group. Water-control group data were represented as 100% (relative to water-control). The values are expressed as mean ± SEM (*n* = 5/group), (** *p* < 0.01).

**Figure 4 biomolecules-10-01030-f004:**
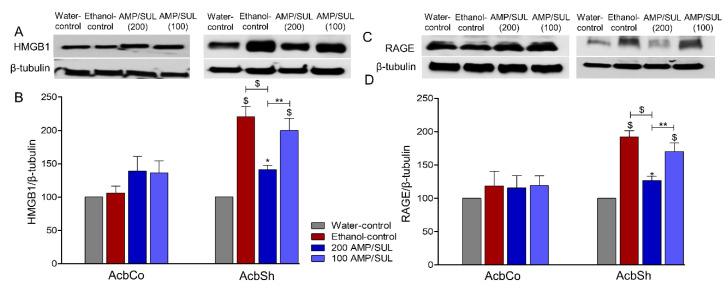
Effects of AMP/SUL treatments (100 mg/kg and 200 mg/kg, i.p.) for five days on the expression of HMGB1 in the AcbCo and AcbSh. (**A**) Immunoblot of HMGB1 and β-tubulin in the AcbCo and AcbSh. (**B**) Quantitative analysis using one-way ANOVA followed by Newman-Keuls test indicated that there were no significant differences in the expression of HMGB1 among the water-control, ethanol-control, ethanol-AMP/SUL (100 mg/kg), and ethanol-AMP/SUL (200 mg/kg) groups in the AcbCo. Quantitative analysis using one-way ANOVA followed by Newman-Keuls test indicated that there was a significant increase in HMGB1 expression in the ethanol-control group compared to water-control group, while post-treatment with AMP/SUL (200 mg/kg, but not 100 mg/kg) for five days normalized HMGB1 expression in the AcbSh. Ethanol-AMP/SUL (100 and 200 mg/kg) groups had higher HMGB1 expression compared to water-control group. (**C**) Immunoblot of RAGE and β-tubulin in the AcbCo and AcbSh. (**D**) Quantitative analysis using one-way ANOVA followed by Newman-Keuls test indicated that there were no significant differences in the expression of RAGE among the water-control, ethanol-control, ethanol-AMP/SUL (100 mg/kg), and ethanol-AMP/SUL (200 mg/kg) groups in the AcbCo. Quantitative analysis using one-way ANOVA followed by Newman-Keuls test indicated that there was a significant increase in RAGE expression in the ethanol-control group compared to the water-control group, while post-treatment with AMP/SUL (200 mg/kg, but not 100 mg/kg) for five days decreased RAGE expression in the AcbSh. Ethanol-AMP/SUL (100 and 200 mg/kg) groups had higher RAGE expression compared to water-control group. The symbol of statistical significance between any group and water-control group is shown on the bar of the group. Water-control group data were represented as 100% (relative to water-control). The values are expressed as mean ± SEM (*n* = 5/group), (* *p* < 0.05, ** *p* < 0.01 and $ *p* < 0.001).

**Figure 5 biomolecules-10-01030-f005:**
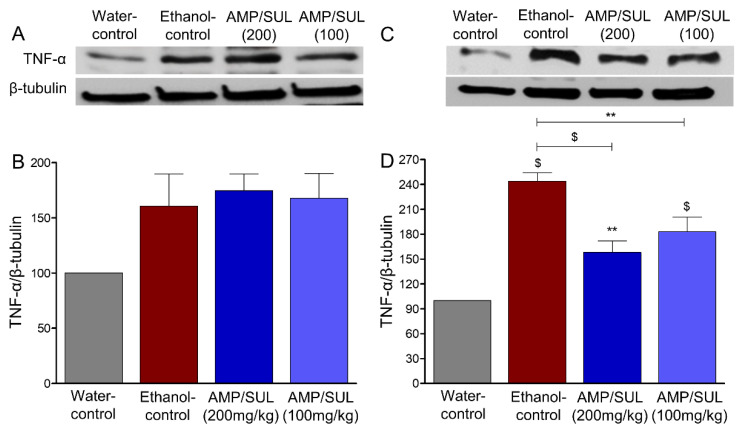
Effects of AMP/SUL treatments (100 mg/kg and 200 mg/kg, i.p.) for five days on the expression of TNF-α in the AcbCo and AcbSh. (**A**) Immunoblot of TNF-α and β-tubulin in the AcbCo. (**B**) Quantitative analysis using one-way ANOVA followed by Newman-Keuls test indicated that there were no significant differences in the expression of TNF-α among water-control, ethanol-control, ethanol-AMP/SUL (100 mg/kg), and ethanol-AMP/SUL (200 mg/kg) rats in the AcbCo. (**C**) Immunoblot of TNF-α and β-tubulin in the AcbSh. (**D**) Quantitative analysis using one-way ANOVA followed by Newman-Keuls test indicated that there was a significant increase in the TNF-α expression in the ethanol control group compared to the water-control group, while post-treatment with AMP/SUL (100 and 200 mg/kg) for five days decreased TNF-α expression in the AcbSh. Ethanol-AMP/SUL (100 and 200 mg/kg) groups had higher TNF-α expression compared to water-control group. The symbol of statistical significance between any group and water-control group is shown on the bar of the group. Water-control group data were represented as 100% (relative to water-control). The values are expressed as mean ± SEM (*n* = 5/group), (** *p* < 0.01 and $ *p* < 0.001).

**Figure 6 biomolecules-10-01030-f006:**
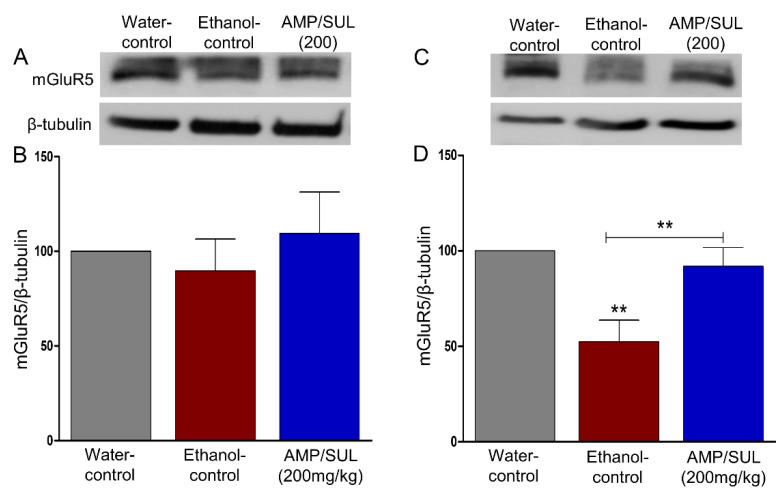
Effects of AMP/SUL treatments (200 mg/kg, i.p.) for five days on the expression of mGluR5 in the AcbCo and AcbSh. (**A**) Immunoblot of mGluR5 and β-tubulin in the AcbCo. (**B**) Quantitative analysis using one-way ANOVA followed by Newman-Keuls test indicated that there were no significant differences in the expression of mGluR5 among water-control, ethanol-control, and ethanol-AMP/SUL (200 mg/kg) groups in the AcbCo. (**C**) Immunoblot of mGluR5 and β-tubulin in the AcbSh. (**D**) Quantitative analysis using one-way ANOVA followed by Newman-Keuls test indicated that there was a significant decrease in the mGluR5 expression in the ethanol-control group compared to the water-control group, while post-treatment with AMP/SUL (200 mg/kg) for five days increased mGluR5 expression in the AcbSh. The symbol of statistical significance between any group and water-control group is shown on the bar of the group. Water-control group data were represented as 100% (relative to water-control). The values are expressed as mean ± SEM (*n* = 5/group), (** *p* < 0.01).

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
