# Peer review of "Ampicillin/Sulbactam Treatment Modulates NMDA Receptor NR2B Subunit and Attenuates Neuroinflammation and Alcohol Intake in Male High Alcohol Drinking Rats"

_biomolecules, 2020, doi:10.3390/biom10071030_

Round 1

Reviewer 1 Report

The revised manuscript by Alasmari et al. addresses my previous concerns without the additional data that was requested.  While that is disappointing, the arguments they present are somewhat supportive of that approach.  While the fact that AMP/SUL alone does not affect place preference does suggests it won’t affect water/food intake or the biomarkers, it would have been nice to actually show that result.  The statement regarding baseline intake is sufficient to indicate appropriate intake in all the rats.  I would put in the text or the figure legend/figure the significant effect of water intake for controls.

Author Response

Reviewer #1:     

The revised manuscript by Alasmari et al. addresses my previous concerns without the additional data that was requested.  While that is disappointing, the arguments they present are somewhat supportive of that approach.  While the fact that AMP/SUL alone does not affect place preference does suggests it won’t affect water/food intake or the biomarkers, it would have been nice to actually show that result.  The statement regarding baseline intake is sufficient to indicate appropriate intake in all the rats.  I would put in the text or the figure legend/figure the significant effect of water intake for controls.

Response:

We thank the reviewer for the comments.  We have added the statistical analysis of water intake within ethanol-control group in Figure 1 legend. 

Reviewer 2 Report

I understand the author's financial hardships and appreciate their efforts to improve the credibility of their conclusion by statistical analysis. but I am reluctant to accept this manuscript without their conclusions backed up by immunohistochemistry. Their conclusions can be different if such other methods are used to study the difference in expression.

Author Response

Reviewer #2:

I understand the author's financial hardships and appreciate their efforts to improve the credibility of their conclusion by statistical analysis. but I am reluctant to accept this manuscript without their conclusions backed up by immunohistochemistry. Their conclusions can be different if such other methods are used to study the difference in expression.

Response:

Although we do respect the opinion of the reviewer #2 about adding immunocytochemistry to this article paper, we do believe that Western blot is an established technique used by most of the scientists to determine accurately changes in protein expression between control and experimental groups. The data generated by Western blot support the conclusion of our paper.

In order to add the immunocytochemistry experiment, the authors will need to have new group of rats (water, saline and AMP/SUL) that will need to be treated similarly as it has been done in our Western blot study. Animals will need to be perfused at the end of the study with PBS and paraformaldeyhyde for brain fixation. To identify changes in the expression of the target proteins, we will need to use stereological microscopic technique. It is unfortunate to note that we don't have the stereology system in our lab to analyze the expression of these proteins using immunocytochemistry in fixed brains. To perform immunocytochemistry, we will need at least 5-7 months of work to complete the study. This will include the time of animal treatments, perfusion, brain sectioning, staining of the selected target proteins using specific antibodies, and the use of stereology system to quantify for the expression of target proteins. In addition, it would be challenging to quantify any stained target proteins that are cytoplasmic versus the ones that are located in the membrane using immunocytochemistry and stereological microscopic system.

In conclusion section, we have added the following: “Moreover, future studies are warranted to investigate the neuronal or cell type involving these excitatory and neuroinflammatory target proteins using immunocytochemistry and stereological system.

Reviewer 3 Report

The authors investigated the effects of ampicillin/sulbactam treatment on the molecular markers of ethanol addiction in the area of nucleus accumbens of ethanol drinking and control rats. They found significant alterations of the expression of glutamate transporter, NR2A, mGluR5, TNFalpha, RAGE and HMGB1 using Western blotting. The antibiotics had advantageous effects as to ethanol dependency and neuroinflammation caused by ethanol consumption. Taken together the article is well-written, original and merits publication.

Further comments on manuscript by Alasmari et al. submitted to Biomolecules.

  1. Introduction: clearly stated effects of chronic ethanol consumption in experimental animals: the rise of extracellular glutamate concentrations and downregulation of GLT-1, based on literature data. The involvement of NR2A and NR2B and mGluR5 are also supported by literature data. The neuroinflammatory background in chronic alcoholics clearly necessitates the investigation of some inflammatory markers: these are HMGB1, RAGE and TNFalpha. We have to note that contrary to the statement of the authors, HMGB1 is not a typical cytokine (but one isotype has cytokine-like actions), because it has a DNA-binding domain. Generally speaking, there are no problems with the Introduction.
  2. Methods: I do not see any problem with the methods although I am not biochemist. The shell and the core of the nucleus accumben can be separated with micropunching. If the authors could insert a microscopic image this could have been more convincing – but I believe this statement without the photograph.
  3. Results: the different data are well-presented. The graphs and the Western blots are convincing. The statistical procedures are appropriate and support the results.
  4. Discussion: the results support the working hypothesis nicely. The arguments and literature data are following each other step-by-step, according to the scheme of the Introduction and the Results. The initial question, i.e. the possible effects of AMP/SUL treatment are supported by the paragraphs of the Discussion. Notes:
  • the discussion should be a little shorter.
  • the possible entry of AMP/SUL into the brain (crossing the BBB) should be discussed. As far as I know these antibiotics can cross the BBB. In this case discussing the gut-brain pathway is not necessary.
  • probably, the authors may consider the insertion of a „Conclusions” section through which they could make the Discussion more straightforward.

Conclusion: I do NOT see major weaknesses in this article. In my opinion, it can be published in the present form. The informations of the article are straightforward and understandable.

Author Response

Reviewer #3:     

Introduction: clearly stated effects of chronic ethanol consumption in experimental animals: the rise of extracellular glutamate concentrations and downregulation of GLT-1, based on literature data. The involvement of NR2A and NR2B and mGluR5 are also supported by literature data. The neuroinflammatory background in chronic alcoholics clearly necessitates the investigation of some inflammatory markers: these are HMGB1, RAGE and TNFalpha. We have to note that contrary to the statement of the authors, HMGB1 is not a typical cytokine (but one isotype has cytokine-like actions), because it has a DNA-binding domain. Generally speaking, there are no problems with the Introduction.

Methods: I do not see any problem with the methods although I am not biochemist. The shell and the core of the nucleus accumben can be separated with micropunching. If the authors could insert a microscopic image this could have been more convincing – but I believe this statement without the photograph.

Results: the different data are well-presented. The graphs and the Western blots are convincing. The statistical procedures are appropriate and support the results.

Discussion: the results support the working hypothesis nicely. The arguments and literature data are following each other step-by-step, according to the scheme of the Introduction and the Results. The initial question, i.e. the possible effects of AMP/SUL treatment are supported by the paragraphs of the Discussion. Notes:

the discussion should be a little shorter.

the possible entry of AMP/SUL into the brain (crossing the BBB) should be discussed. As far as I know these antibiotics can cross the BBB. In this case discussing the gut-brain pathway is not necessary.

Probably, the authors may consider the insertion of a „Conclusions” section through which they could make the Discussion more straightforward.

Conclusion: I do NOT see major weaknesses in this article. In my opinion, it can be published in the present form. The informations of the article are straightforward and understandable.

Response:

We thank the reviewer for the positive comments.   We have changed “an inflammatory cytokine” to “an inflammatory mediator” with HMGB1.  We have discussed the penetration of ampicillin and sulbactam into the brain as suggested.  We have clarified the use of micropunching procedure to isolate the brain regions.  In addition, we have shorten the discussion section as suggested by the reviewer.  We also have inserted a conclusion section as suggested.

Round 2

Reviewer 2 Report

I understand the author's hardships, but it does not justify the lack of supporting evidence in addition to Western blotting to conclude the difference between AcbCo and AcbSh. Immunohistochemistry of at least some of the Western blotted proteins is critical to see whether the dissection of the two regions was performed properly by the authors.

This manuscript is a resubmission of an earlier submission. The following is a list of the peer review reports and author responses from that submission.

Round 1

Reviewer 1 Report

The manuscript by Alasmari et al. reports on the effects of treatment with AMP/SUL following chronic ethanol exposure.  They find that the treatment decreases ethanol intake in the last week of exposure, while simultaneously increasing water intake.  A small decrease in food intake was noted.  Chronic ethanol also decreases GLT-1 in the acb shell and NR2B in the core.  Ethanol increases the neuro-inflammation factors HMGB1, RAGE and TNF-alpha in the shell.  All of these effects were reversed by AMP/SUL treatment.  Overall, the findings are certainly interesting and should be relevant to research on alcohol abuse.  I do have some concerns.

My major concern is a lack of an AMP/SUL control in the absence of ethanol treatment.  We don’t know the effects of the treatment alone, making it difficult to determine the true mechanism of the effect.  If the treatment alone has effects in the opposite direction of ethanol, then the treatment may just counteract the ethanol effect in a simple additive manner.  If this were the case, it would still be potentially important as a treatment, but it would have implications for further research.

I would also like to see the ethanol/intake data from the first 5 weeks of treatment.  Do the rats drink the higher concentration of ethanol?

Water intake appears to increase for the water control group in Figure 1.  Is that effect significant?  Why might it be happening?

Does the loss of the small effect on food intake represent tolerance to the drug?

There are a few instances where the English could be improved.

Reviewer 2 Report

This is a study about the effect of b-lactam and lactamase inhibitor on ethanol drinking behavior neuroinflammation in nucleus accumbens. Relationships between b-lactams, glutamatergic systems, addiction, ethanol, and brain regions have been studied extensively, as shown in the introduction. This is a study to evaluate the effect of ampicillin/sulbactam, focused on the difference of the core and shell of the nucleus accumbens.

The most critical issue is that, although NR2B expression level reached statistical significance in AcbCo but not in AcbSh, some reduction in NR2B similar to that of AcbCo is observed in the graph. This is at the core of this study, claiming novelty, and mentioned many times in the discussion. A more careful analysis of NR2B, increasing the sample number, immunohistochemical analysis, and statistical comparison between AcbCo and AcbSh is required for this conclusion.

(3.3, 3.4, Figure 4)

To provide more statistical information, it will be better to show the data points of each repeated experiment (n) for all graphs.

Immunohistochemistry microscope images of nucleus accumbens for at least some molecules used in this study should be provided.

Theory and practical advantages and disadvantages of using AMP/SUL, instead of using AMP only should be discussed.

The following are minor issues that caught my attention:

L14 Neuroinflammation is a common manifestation associated with chronic exposure to ethanol. 

-> Exposure to ethanol commonly manifests neuroinflammation. 

L33 Previous revealed -> Our previous studies revealed 

L63 HMGB1 appears here first before abbreviation is shown in L65

L76 Postmortem -> postmortem

L77 HMGB1is -> HMGB1 is

L82 may induces -> may induce

L85 Knowing that ~ ; incomplete sentence.

L89  HAD1rats ; needs definition